# TRANSFORMER TRAINING INSTABILITY OF SOFTMAX AND LIPSCHITZ KERNEL ATTENTIONS

## ABSTRACT

Transformers have been making significant progress across various domains, and recently, with scaling up of models like LLMs, they have achieved even greater success. Recent findings have shown that the softmax function in the self-attention used to re-weight the attention logits into probability vectors causes *attention entropy collapse*, where the attention is concentrated on a single token, and it leads to unstable training. In this work, we first demonstrate that the (non-Lipschitz) softmax-based attention leads to the attention entropy collapse but the *Lipschitz-kernel*-based attention does not. We show that the Lipschitzness of the attention plays an important role in keeping the attention entropy stable regardless of the variance of the attention logits. Moreover, we argue that the underlying reason why the attention entropy collapse leads to the training instability is that as the attention probabilities become more concentrated, it causes the attention matrix to gradually increase, leading to gradient exploding.

## 1 INTRODUCTION

Since Transformer (Vaswani et al., 2017) was introduced in the field of Natural Language Processing (NLP), it has recently demonstrated strong performance in various areas, including computer vision (Dosovitskiy et al., 2021), speech (Dong et al., 2018), and other fields as well. A key factor contributing to the success of Transformers across diverse fields is their capacity to effectively capture long-range dependencies via the self-attention mechanism. Specifically, the softmax function transforms attention logits into probability distributions, reflecting the relative importance of each element based on its relevance to the others. This mechanism facilitates the remarkable performance of Transformers, normalizing large values impact on the weighted sum and effectively emphasizing relative importance. However, this can lead to *attention entropy collapse*, negatively impacting the model's performance and training stability.

Attention entropy collapse arises when the query-key inner product (also called the attention logit) is converted into a probability vector, leading to an excessive focus on a single token. As a result of attention entropy collapse, the model struggles to effectively learn the relationships between inputs, resulting in degraded performance (Zhai et al., 2023). Entropy collapse as being caused by the softmax function, which exponentially magnifies the differences between query-key inner products when re-weighting them into probabilities (Wang et al., 2021; Shen et al., 2023). However, it is challenging to identify the reason why the softmax function is prone to attention entropy collapse and why entropy collapse causes instability in training.

In this work, we demonstrate that the (non-Lipschitz) softmax-based attention leads to the attention entropy collapse but the *Lipschitz-kernel*-based attention does not. In other words, the Lipschitzness of the attention plays an important role in keeping the attention entropy stable. Figure 1 shows that softmax-based self-attention and its variant applying Layer Normalization (Gilmer et al., 2023; Dehghani et al., 2023) lead to entropy collapse, but Lipschitz kernel self-attention (ReLU, ELU+1) ensures stable training. Additionally, as the entropy collapses to zero, the gradient explodes, and the loss diverges. Previous approaches primarily focused on interpreting the results; however, we emphasize bridge that, during entropy collapse, when attention probabilities take the form of one-hot vectors, the norm of the corresponding matrix increases, leading to gradient exploding.

First, softmax-based self-attention is more susceptible to entropy collapse compared to Lipschitz-kernel-based self-attention due to its increased sensitivity to variance in attention logits. We conduct

controlled experiments to examine how both functions re-weight into different types of probability distributions when the variance of the attention logits increases. As variance increased, softmax produced highly imbalanced distributions, while the Lipschitz kernel maintained stable entropy. We also observed that as the variance of softmax inputs grows during training (Jiang et al., 2023), the distribution becomes more concentrated, resulting in a one-hot-like vector. In contrast, the Lipschitz kernel stabilizes weight distribution despite high variance in logits. Therefore, compared to the Lipschitz kernel function, softmax, which is highly sensitive to the variance of attention logits, tends to increase this sensitivity during training, leading to attention entropy collapse.

In addition, as shown in Figure 1, as attention entropy reaches zero during training, the gradient norm explodes, preventing model convergence. It is crucial to analyze how attention entropy collapse contributes to these instability issue. The softmax function outputs exponentially larger values in response to its inputs, making it not Lipschitz continuous, and as a result, self-attention is not Lipschitz, which can lead to very large gradients (Kim et al., 2021; Dasoulas et al., 2021). It is noted that when the attention probabilities distribution becomes highly concentrated, the norm of attention probabilities matrix increases, causing larger gradients in the backward pass and leading to exploding gradients. Through experiments, we observe that softmax-based self-attention increases attention probabilities norms, leading to exploding gradients, while Lipschitz kernel self-attention keeps norms and gradients stable due to lower function variation.

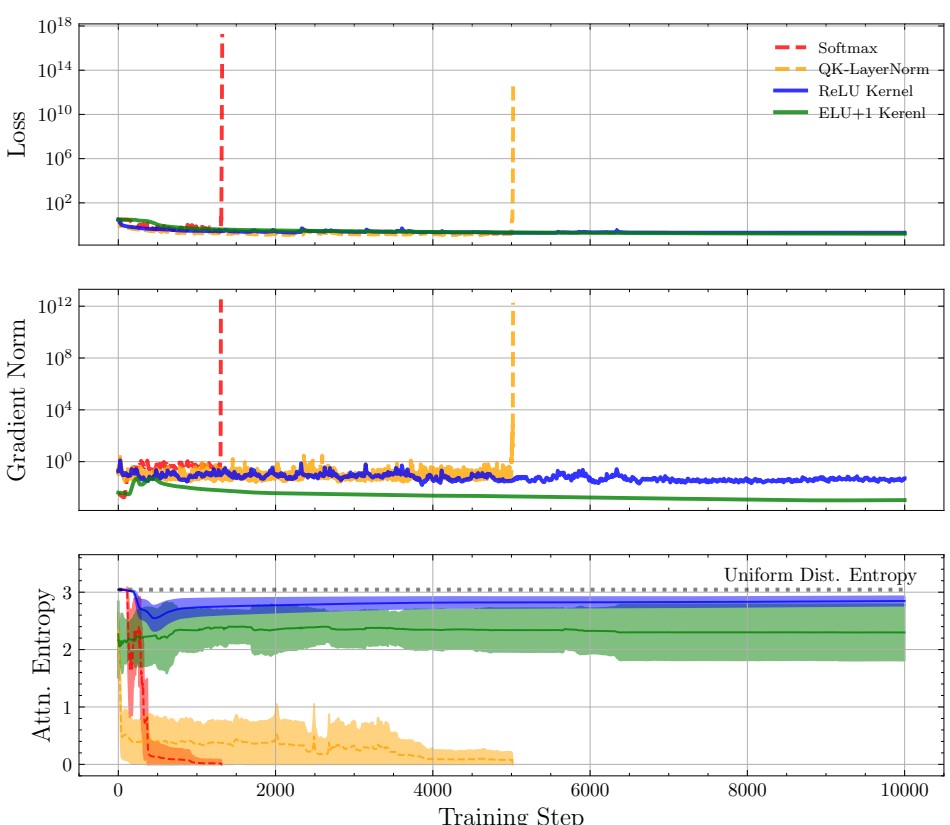

Figure 1: Softmax-based self-attentions (dashed lines; Softmax, QK-LayerNorm) exhibit instability due to decreasing attention entropy, whereas Lipschitz kernel self-attentions (solid lines; ReLU, ELU+1) demonstrate stable training with entropy hovering away from 0. The attention entropy is the average entropy in (6) of the first self-attention layer (see Appendix D for the other layers). We also indicate the maximum entropy in the case that the attention probability is uniform (dotted line).

## 2 BACKGROUND

### 2.1 SOFTMAX BASED SELF-ATTENTION

Given an input $X \in \mathbb{R}^{N \times D}$ with sequence length $N$ and hidden dimension $D$. We can denote three components of single head self-attention, query $Q \in \mathbb{R}^{N \times D}$, key $K \in \mathbb{R}^{N \times D}$, value $V \in \mathbb{R}^{N \times D}$ by multiplying each corresponding weight $W_Q, W_K, W_V \in \mathbb{R}^{D \times D}$ with $X$. Then, the $i$th row vector $A_i \in \mathbb{R}^{1 \times D}$ of self-attention's output $A \in \mathbb{R}^{N \times D}$ and $i$th and $j$th elements of the attention (probability) matrix $P \in \mathbb{R}^{N \times N}$ can be defined as follows:

$$A_i = \sum_{j=1}^{N} P_{i,j} V_j, \text{ where } P_{i,j} = \frac{\text{sim}(Q_i, K_j)}{\sum_{j=1}^{N} \text{sim}(Q_i, K_j)} \tag{1}$$

and $\text{sim}(\cdot)$ is a real-valued function that measures the similarity between query and key. In the softmax-based self-attention, the query-key inner product $Z_{i,j} = Q_i K_j^\top$ with the exponential function is used for this purpose as follows:

$$P_{i,j} = \frac{\exp(Q_i K_j^\top)}{\sum_{j=1}^{N} \exp(Q_i K_j^\top)}, \tag{2}$$

$$\text{sim}(Q_i, K_j) = \exp(Z_{i,j}), \text{ where } Z_{i,j} = Q_i K_j^\top. \tag{3}$$

Note that, because of the dot-product $QK^\top$ with $Q$ and $K$, this operation has quadratic complexity $O(N^2 D)$ with respect to input sequence length $N$.

### 2.2 LIPSCHITZ KERNEL SELF-ATTENTION

The computational cost of softmax based self-attention has been an issue, because the computational cost increases quadratically with the sequence length. Thus, to overcome this limitation, various approaches have been proposed, such as sparse pattern (Beltagy et al., 2020; Zaheer et al., 2020), low-rank (Wang et al., 2020; Hu et al., 2021) and kernelized self-attention (Choromanski et al., 2020; Cai et al., 2022). Among them, kernelized self-attention approximate the similarity function by using kernel function $\phi : \mathbb{R}^{1 \times D} \to \mathbb{R}^{1 \times D}$ as follows:

$$\text{sim}(Q_i, K_j) \approx \phi(Q_i)\phi(K_j)^\top. \tag{4}$$

With this similarity function, we can rewrite 1 with associativity property of matrix products to reduce quadratic complexity into linear complexity as follows:

$$A_i' = \frac{\phi(Q_i) \sum_{j=1}^{N} \phi(K_j)^\top V_j}{\phi(Q_i) \sum_{j=1}^{N} \phi(K_j)^\top}. \tag{5}$$

Based on (5), we can calculate $\phi(K)^\top V \in \mathbb{R}^{D \times D}$ instead of query key dot-product, $QK^\top \in \mathbb{R}^{N \times N}$. As a result, we can reduce quadratic time complexity into $O(ND^2) = O(N)$ with respect to the input sequence length $N$ as the hidden dimension is smaller than sequence length. The primary direction of works in kernelized self-attention has focused on how to approximate softmax function with a kernel function $\phi$. To approximate the softmax function, previous approaches reflect its characteristic typically $P$ can only consist of positive values by replacing $\phi$ with ReLU and re-weighting function (Qin et al., 2022; Cai et al., 2022; Han et al., 2023), ELU+1 (Katharopoulos et al., 2020), and many others (Chen et al., 2021; Arora et al., 2024; Aksenov et al., 2024; Zhang et al., 2024).

### 2.3 ATTENTION ENTROPY

The entropy of the attention probability $P$, also called *attention entropy*, can be computed by calculating the entropy for each row $P_i$ and taking the average, as follows:

$$H(P) = \frac{1}{N} \sum_{i=1}^{N} \left( -\sum_{j=1}^{N} P_{i,j} \log(P_{i,j}) \right). \tag{6}$$

If the probability distribution becomes skewed, with most of the probability mass concentrated on a single token, the entropy gradually decreases. This issue is referred to as *attention entropy collapse*. This issue similarly appears in LLMs, where, for example, a few activation units exhibit significantly larger values than others, and these large activations lead to the concentration of attention probability on their corresponding tokens (Sun et al., 2024). Additionally, entropy collapse, where attention concentrates on a few tokens, is linked to grokking, where performance suddenly improves after a long period of LLMs performance (Hoffmann et al., 2023). There are several approaches aimed at addressing the problems of entropy collapse.

**Normalization**   One of the causes of attention entropy collapse is that when the magnitude of attention logits is large for a specific token or a small set of tokens, the softmax output becomes sharply peaked, focusing almost all attention on those few tokens while others receive nearly zero. A representative method to mitigate this issue is QK-LayerNorm (Gilmer et al., 2023; Dehghani et al., 2023), which applies layer normalization to both the query and key before passing them into the softmax function, effectively reducing their magnitudes. To mitigate the issue caused by significant statistical variations in input values, NormSoftmax (Jiang et al., 2023) normalizes the input vectors to the softmax function, ensuring they have unit variance. Also, $\sigma$Reparam prove that as the $\ell_2$-norm of the product of the query and key weights increases, the bound becomes tighter, resulting in low entropy. By reparameterizing the parameters using spectral normalization, it was possible to prevent the entropy from reaching zero.

**Activation Function**   There are also approaches that address the entropy collapse issue by modifying the softmax function from a different perspective. ReLUFormer (Shen et al., 2023), by replacing the exponential function in softmax with ReLU, they resolved the issue of insufficient key-value memory slots caused by the centralized softmax function. Additionally, in Wang et al. (2021) the attention scores between embedding vectors are not normally distributed because of softmax function, so the exponential function is replaced with a periodic function e.g sine, cosine to resolve the training instability. In this work, we focus on the variance of attention logits, the most sensitive factor causing entropy collapse in softmax-based self-attention, and reveal that the Lipschitz kernel function, a simple method without parameters, is highly robust to entropy collapse.

## 3 Entropy and Instability between Softmax-based and Lipschitz kernel self-attention

### 3.1 Setup

Recent studies have reproduced some significant behaviors of large scale models using smaller proxy models, effectively capturing these behaviors with improved computational efficiency (Wortsman et al., 2023; Tan et al., 2024). This allows for valuable insights into core dynamics with a small computational costs. By using toy models, one can effectively explore grokking (Power et al., 2022; Liu et al., 2022), where a model suddenly improves its generalization and performance on unseen data. Also, with small proxy models, it is possible to replicate the divergence issues between attention logit and output logit that occur in large models (Wortsman et al., 2023). Thus, we specify the experimental setup based on these works.

**In-Context Linear Regression**   Recently, it has been shown that Transformer has an emergent behavior, called in-context learning (Brown et al., 2020), that it can learn (without updating any parameters) a new unseen task from a few demonstrations of input-output pairs, leading to extensive research on Transformer training (Garg et al., 2022; Zhang et al., 2023; Mahankali et al., 2023; Von Oswald et al., 2023; Ahn et al., 2023; 2024). The in-context linear regression task involves pairs $\{(x_i, y_i)\}_{i=1}^n$ with the query vector $x_{n+1}$, where $x_i$'s and $w$ are drawn i.i.d. $\mathcal{N}(0, I_D)$ and $y_i = w^\top x_i$. The objective of this task is to train a Transformer $T_\theta$ to take $(x_{1:n+1}, y_{1:n})$ as inputs and predict $w^\top x_{n+1}$ with the following loss:

$$\mathcal{L}(\theta) = \frac{1}{2}\mathbb{E}_{x_{1:n+1}, w}\left[\left(T_\theta(x_{1:n+1}, y_{1:n}) - w^\top x_{n+1}\right)^2\right]. \quad (7)$$

Note that the input sequence length $N$ satisfies $N = n + 1$ as we are given $n$ demonstrations $\{(x_i, y_i)\}_{i=1}^n$ along with a query vector $x_{n+1}$.

**Only Self-attention layer** We use a Transformer with a stack of self-attention layers. By using only the self-attention layer as a model component, it becomes easier to analyze the training process when attention entropy collapse occurs. Additionally, this allows for focused monitoring of how such limitations impact optimization and contribute to learning instability. This approach helps isolate the influence of self-attention and provides a detailed understanding of how entropy collapse affects model performance.

**Lipschitz Kernel Function** The softmax function in self-attention is not Lipschitz continuous because it amplifies small input differences exponentially, leading to unbounded output changes (Dasoulas et al., 2021; Kim et al., 2021). To compare with the softmax-based attention, we experiment with a kernelized self-attention with a Lipschitz kernel function (i.e., for some $\alpha > 0$, $\|\phi(x) - \phi(x')\| \leq \alpha\|x - x'\|$ for any $x, x'$), *Lipschitz kernel self-attention*. which we expect to mitigate entropy collapse. To be specific, we use ReLU and ELU+1 as kernel functions, which ensure non-negative values, and we demonstrate that these functions are are less sensitive to attention entropy collapse.

**Definition 3.1** (Lipschitz Kernel Attention). A kernelized attention in (5) is called *Lipschitz kernel attention* when the kernel function $\phi$ is Lipschitz, i.e., there is a constant $\alpha > 0$ such that, for any $x, x'$,

$$\|\phi(x) - \phi(x')\| \leq \alpha\|x - x'\|.$$

Specifically, we use ReLU (Qin et al., 2022; Cai et al., 2022; Han et al., 2023) and ELU+1 (Katharopoulos et al., 2020), both simple and commonly used Lipschitz kernel functions with the Lipschitz constant $\alpha = 1$. Additionally, while kernelized methods offer the benefit of reduced computation, computing the $N \times N$ attention probabilities to obtain the entropy still requires evaluating all query-key pairs. Therefore, in this case, we did not pre-compute the inner product of query and key, but instead followed the traditional approach, calculating each query-key interaction individually.

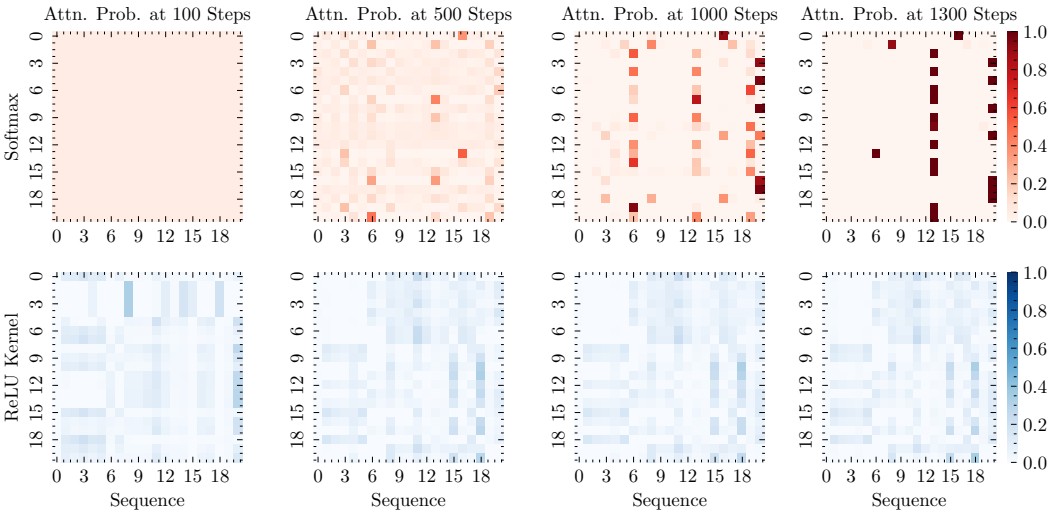

Figure 2: Changes in attention probabilities between softmax-based (Top) and Lipschitz kernel self-attention (Bottom) during training. In softmax-based self-attention, each row progressively converges toward a one-hot vector, eventually resulting in complete one-hot vectors, which leads to attention entropy collapse. The attention matrices are from the first layer.

### 3.2 ATTENTION ENTROPY BEHAVIOUR BASED ON THE RE-WEIGHTING FUNCTION

We find that the softmax function is susceptible to entropy collapse, but when approximated with a Lipschitz kernel, it remains robust against such collapse. Additionally, due to this collapse, the

softmax-based methods show that gradient norm increases rapidly, causing the loss to diverge and making the learning process unstable. Figure 1 shows that, in the case of self-attention using the softmax function, the entropy gradually decreases in the early steps, eventually reaching zero, while the gradient increases sharply. This confirms that entropy collapse makes the model's learning unstable, and unlike the gradient vanishing observed in large models, the gradient is found to explode. As attention entropy reaches zero, it indicates that the attention probabilities in each layer have transformed into one-hot vectors, as shown in Figure 2 (Top). Furthermore, even after applying QK-LayerNorm (Gilmer et al., 2023; Dehghani et al., 2023), a representative method to prevent the exponential growth of attention logits, it still results in unstable learning as entropy collapses. This is inferred to be caused by the inner product of the individually normalized query and key being re-weighted into an extreme probability distribution through the softmax function. In contrast, in the case of Lipschitz kernel self-attention, it is observed that the model maintains a more stable and higher entropy level compared to the softmax-based attentions, leading to more stable training.

**Why is Softmax Vulnerable to Entropy Collapse Compared to Lipschitz Kernels**  The main difference between the softmax function and Lipschitz kernel functions lies in their handling of input sensitivity and output stability (Kim et al., 2021). Due to its exponential nature, softmax function is highly sensitive to small variations in input. As the variance between values increases, the exponentiation leads to increasingly larger differences. This can cause entropy collapse, where attention becomes overly concentrated on a few tokens, reducing diversity and destabilizing the training process. In contrast, Lipschitz kernel functions maintain a controlled, proportional relationship between input and output changes, ensuring that the attention distribution remains balanced and stable. Theoretically, this stability arises because Lipschitz kernel functions have bounded gradients, which prevent the gradients from exploding or vanishing during training. By limiting the rate at which the output can change relative to the input (as defined by the Lipschitz constant), these functions promote smoother training and help maintain a diverse and effective attention mechanism. Thus, in self-attention, the softmax function distorts the probability distribution, concentrating most values in one area and creating positive skewness. In contrast, the Lipschitz kernel function limits changes, leading to a more evenly spread distribution with lower skewness. The detailed description regarding experiment on the difference in skewness between the two functions is in the Appendix C.

### 3.3 ENTROPY IS SENSITIVE TO VARIANCE FOR THE SOFTMAX-BASED SELF-ATTENTION

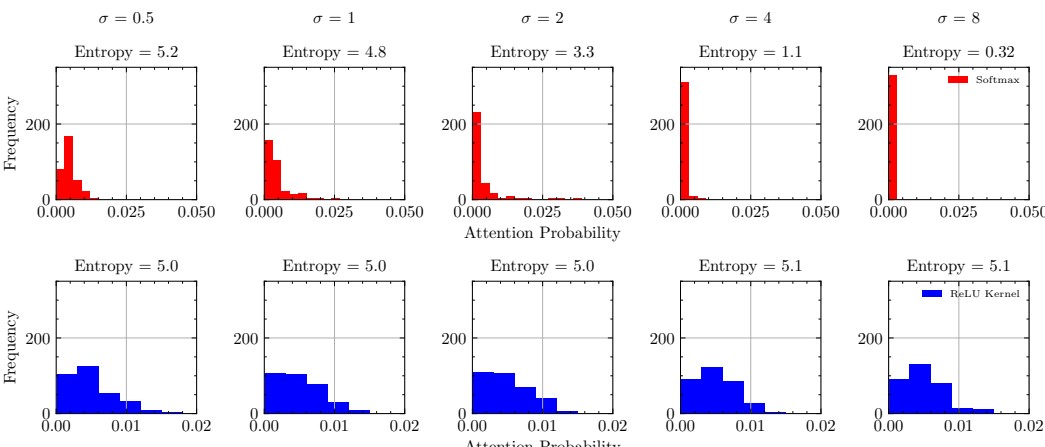

Figure 3: Comparison of the distributions of attention entropy between softmax-based self-attention (Top) and Lipschitz kernel self-attention (Bottom) as the attention logit variance increases without learning. With query norm fixed at 1 and keys sampled with increasing variance $\sigma^2$, softmax-based self-attention shows skewed distributions and low entropy as logit variance increased, while Lipschitz kernel self-attention remains stable.

In this section, we analyze why self-attention with the Lipschitz kernel is more stable in terms of entropy than the softmax-based attention. Softmax exponentially amplifies the relative differences

between input values, making larger differences more pronounced. In contrast, the Lipschitz kernel function has the property of linearly limiting the changes in response to variations in the input values. When the differences between input values increase, the Lipschitz kernel function does not significantly amplify those differences and maintains stability. Therefore, the case where the softmax function becomes highly vulnerable to entropy collapse is when the variance of the inner product between the query and key increases. We call the variance of inner product between query and key *the attention logit variance* and define it as:

**Definition 3.2** (Attention Logit Variance). The attention logit variance for each row $Z_i$ of the attention logit $Z \in \mathbb{R}^{N \times N}$ is defined as follows:

$$\text{Var}_j[Z_{i,j}] = \frac{1}{N} \sum_{j=1}^{N} \left( Z_{i,j} - \frac{1}{N} \sum_{k=1}^{N} Z_{i,k} \right)^2. \tag{8}$$

**Controlled Experiment** We conduct a controlled experiment to examine the sensitivity of softmax-based and Lipschitz kernel self-attention to the attention logit variance. First, to control the variance of the inner product between the query and the keys (attention logit variance), we fix the query as a unit vector of length 1 and sample the keys from $\mathcal{N}(0, \sigma^2 I)$. Figure 3 shows the histograms of entropy for different $\sigma = 0.5, 1, 2, 4, 8$, i.e., different attention logit variances, for the softmax-based and Lipschitz kernel attentions, respectively. Applying softmax to these inner products will lead to more extreme distributions, concentrating higher probabilities on a few key vectors. This results in the distribution becoming more peaked, with most of the probability mass concentrated on a few values. Consequently, the entropy of the distribution decreases significantly as the variance of the key vectors increases, confirming that the softmax output becomes increasingly skewed and less diverse. In contrary, in the case of Lipschitz kernel self-attention, even as the variance increases, the attention distribution does not become as extreme, and the entropy remains more stable. This is because kernel-based methods often use functions that are less sensitive to large disparities in the input values compared to softmax. As a result, the probability distribution remains more balanced, avoiding the sharp concentration of probabilities that can occur with softmax, thus preventing entropy collapse and maintaining a more evenly spread attention distribution. The correlation between variance and entropy for a normal distribution is detailed in the Appendix B.

## 3.4 ENTROPY COLLAPSE WITH INCREASING VARIANCE IN SOFTMAX DURING TRAINING

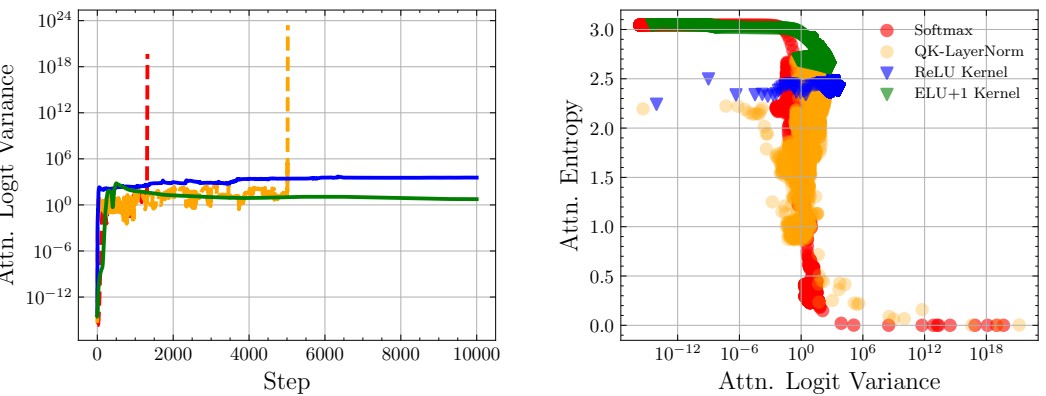

Figure 4: Changes in the attention logit variance and its correlation with attention entropy during training. (Left) For softmax-based self-attention, the attention logit variance grows abruptly, while in Lipschitz kernel self-attention, the variance remains stable. (Right) It shows the relationship between attention entropy and attention logit variance, where entropy stays constant in Lipschitz kernel self-attention but drops significantly in softmax-based self-attention as the attention logit variance increases, showing a clearly inverse relationship.

Through controlled experiments, it was observed that as variance increases, softmax reacts much more sensitively than the Lipschitz kernel, with entropy decreasing sharply, potentially leading to en-

tropy collapse. Therefore, the analysis explores the variance of attention logits throughout the training process of both softmax-based self-attention and Lipschitz kernel self-attention mechanisms, as well as the relationship between variance and entropy.

First, the focus is on how the attention logit variance changes during training with softmax-based self-attention and Lipschitz kernel self-attention. Analyzing the attention logit variance helps identify significant disparities that may lead to extreme attention distributions in softmax-based mechanisms. As illustrated in Figure 4 (Left), self-attention layers based on softmax show increasing variance in the inner product between queries and keys throughout training, while self-attention using ReLU as the kernel initially increases in variance but then stabilizes. The rising variance can lead to entropy collapse through the softmax function, potentially limiting model performance. In contrast, kernelized self-attention maintains variance after passing through the re-weighting function, contributing to more stable training dynamics.

In addition, since the magnitude of the variance in the attention logits is directly linked to entropy collapse during training, the experiment also examines how the variance of the attention logits changes in softmax-based self-attention and Lipschitz kernel self-attention. From right figure illustrates the correlation between attention logit variance and corresponding entropy during training. In softmax-based self-attention, increased variance results in decreased entropy, demonstrating an inverse relationship, while Lipschitz kernel self-attention maintains stable entropy. Even at the same variance, softmax shows lower entropy, indicating that it leads to extreme probability distributions that can compromise training stability. In contrast, Lipschitz kernel self-attention retains a more stable entropy, highlighting the distinct relationship between variance and entropy in these mechanisms.

In conclusion, softmax-based self-attention is highly sensitive to attention logit variance, with sharp increases during training that make it prone to entropy collapse. In contrast, Lipschitz kernel self-attention keeps variance and distribution differences more stable.

## 3.5 WHY DOES ENTROPY COLLAPSE LEAD TO UNSTABLE TRAINING INSTABILITY

The instability caused by entropy collapse during training is a well-known issue, with several research trying to address the issue (Wang et al., 2021; Wortsman et al., 2023; Zhai et al., 2023; Dehghani et al., 2023). Figure 1 shows that, the gradient explodes, leading to a corresponding spike in loss, rendering the model unable to continue training. The sharp growth in gradient norms during entropy collapse has been observed, but the specific mechanisms behind the gradient explosion are unclear. Since the model is composed solely of self-attention layers, it is free from the influence of gradient operations arising from other components.

**Exploding Gradients and Entropy-Collapsed Attention Probabilities**  Softmax-based self-attention can lead to training instability because the softmax function is not Lipschitz continuous, and this sensitivity to changes in the input can result in gradient exploding. In specific, the derivative of the softmax-based self-attention output with respect to the input increases, which is related to the attention probabilities matrix. According to Dasoulas et al. (2021), the norm of the derivative of self-attention layer with respect to the input $X$ can be upper bounded as follows:

$$\|\mathbf{D}A_X\|_{F,F} \leq \|P\|_F + \sqrt{2}\|X\|_{(2,\infty)} \|\mathbf{D}Z_X\|_{F,(2,\infty)}, \tag{9}$$

where $\|X\|_{(2,\infty)} = \max_j (\sum_i X_{i,j}^2)^{1/2}$ and $\|f\|_{a,b} = \max_{\|x\|_b=1} \|f(x)\|_a$.

In (9), the first attention probability term $\|P\|_F$ is related to attention entropy. When the entropy approaches zero—meaning the attention becomes highly extreme and concentrates on a single element—the Frobenius norm of the attention probabilities increases to $\|P\|_F = \sqrt{\sum_{i,j=1}^N P_{i,j}^2} = \sqrt{\sum_{i=1}^N 1} = \sqrt{N}$ because the sum of the squares of the one-hot probabilities is 1. On the other hand, when the attention probabilities are uniformly distributed, each element $P_{i,j}$ is $1/N$ leading the sum of the squares to be $\sum_j 1/N^2 = 1/N$ and resulting in a smaller norm of $\|P\|_F = 1$. This illustrates that as the attention distribution becomes more concentrated, the norm of the attention probabilities increases, whereas a uniform distribution yields a smaller norm of attention matrix. Therefore, the more extreme the distribution of attention probabilities is, the larger the gradient becomes. This

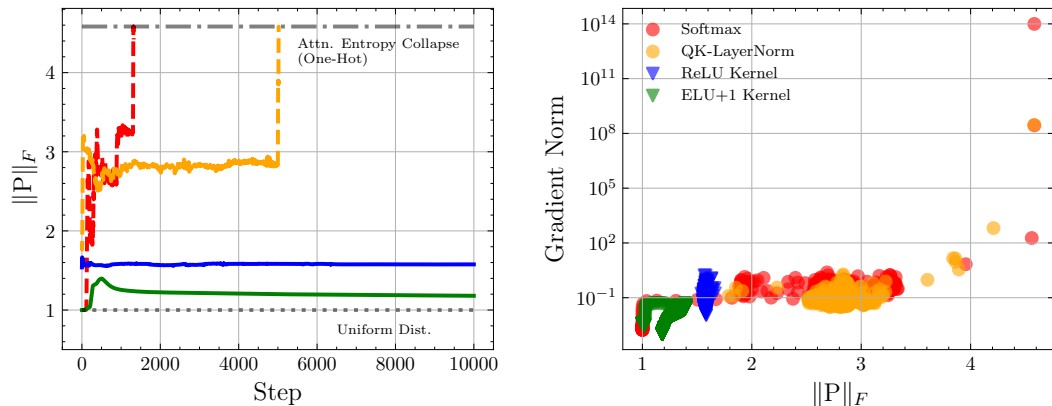

Figure 5: Growth of attention probabilities norms, $\|P\|_F$, and their correlation with gradient in softmax-based and Lipschitz kernel self-attention. (Left) For softmax-based self-attention, $\|P\|_F$ increases during training, leading to one-hot attention probabilities (dashdotted line; $\|P\|_F \approx \sqrt{N}$) and unstable training. In contrast, Lipschitz kernel self-attentions show $\|P\|_F \approx 1$ which corresponds to a uniform distribution (dotted line). (Right) $\|P\|_F$ and the gradient are correlated, leading to unstable learning in softmax-based self-attention as $\|P\|_F$ increases sharply.

behavior highlights the sensitivity of the model to input changes, emphasizing the importance of managing the distribution of attention probabilities to maintain stable training.

## 4 CONCLUSION

In this paper, we reproduced the instability of Transformer training and entropy collapse in small-scale models and found that this instability is heavily influenced by the attention method used to re-weight the attention logits. Through experiments, we confirm that Lipschitz kernel self-attentions maintain higher and more stable entropy during training compare to the softmax-based attentions. Our theoretical and experimental results demonstrate that the softmax function, as the attention logit variance increases, re-weights the logits into more extreme distributions, eventually resembling one-hot vectors. Additionally, our experiments show that, compared to the Lipschitz kernel attentions, the attention logit variance in the softmax attention increases more during training, making it more prone to entropy collapse. Furthermore, as the attention matrix becomes sparse, its norm increases together with the gradient, leading to training instability.

## ETHICS STATEMENT

This paper presents work whose goal is to advance the field of Machine Learning. There are many potential societal consequences of our work, none which we feel must be specifically highlighted here.

## REPRODUCIBILITY STATEMENT

Detailed explanations of the experimental settings are provided in Appendix A. We also attach our code to facilitate the reproduction of our experiments.

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

## A  IMPLEMENTATION DETAILS

Here are the hyper-parameters we used, and we apply the same ones across all experiments.

Table 1: Hyper-parameters of a Simple Transformer

| Parameter | Value |
|---|---|
| Optimizer | SGD |
| Momentum | 0.8 |
| Learning rate | 0.5 |
| Hidden dimension | 3 |
| Sequence length | 20 |
| Attention heads | 1 |
| Attention layers | 5 |
| Training Step | 10000 |
| Gradient Clipping | 1000 |

We focused on how entropy behaves when using different functions in self-attention, and in order to approximate entropy collapse in large models and reproduce it in smaller models, we set a high learning rate of 0.5. Since there was no significant difference between the SGD and Adam optimizers, we used the SGD optimizer. We configured the model with a batch size of 4000, and due to the small model size, we set the number of layers to 5, the number of heads to 1, the sequence length to 20, and the hidden dimension to 5. To investigate the nature of the gradient without constraints, we excluded gradient clipping, and all training was conducted over 10,000 steps.

## B  PROOF OF CORRELATION BETWEEN VARIANCE AND ENTROPY

If the distribution follows a normal distribution, we can define probability density function (PDF) of the normal distribution $X \sim N(\mu, \sigma^2)$ for observation $x$:

$$g(x) = \frac{1}{\sqrt{2\pi\sigma^2}} \exp\left(-\frac{(x-\mu)^2}{2\sigma}\right) \tag{10}$$

where $\mu$ is the mean and $\sigma^2$ is the variance of distribution.

Also, we can define entropy of $X$ as:

$$H(X) = -\int_{-\infty}^{\infty} g(x) \log g(x) dx \tag{11}$$

To compute logarithm of $g(x)$, we can use properties of it:

$$\log g(x) = \log\left(-\frac{1}{2\pi\sigma^2}\right) + \log\left(\exp\left(-\frac{(x-\mu)^2}{2\sigma^2}\right)\right) \tag{12}$$

$$\log g(x) = -\frac{1}{2}\log(2\pi\sigma^2) - \frac{(x-\mu)^2}{2\sigma^2} \tag{13}$$

Then, we can calculate $H(X)$ with replacing $\log g(x)$ with (13):

$$H(X) = -\int_{-\infty}^{\infty} g(x)\left(-\frac{1}{2}\log(2\pi\sigma^2) - \frac{(x-\mu)^2}{2\sigma^2}\right)dx \tag{14}$$

We can separate two terms and first term can be computed using $-\int_{-\infty}^{\infty} g(x)dx = 1$:

$$-\int_{-\infty}^{\infty} g(x) \left(-\frac{1}{2}\log(2\pi\sigma^2)\right) = \frac{1}{2}\log(2\pi\sigma^2) \tag{15}$$

In normal distribution, with $-\int_{-\infty}^{\infty} f(x)(x-\mu)^2 dx = \sigma^2$ we can simplify the second term as:

$$-\frac{1}{2\sigma^2}\int_{-\infty}^{\infty} g(x)(x-\mu)^2 dx = -\frac{1}{2\sigma^2}\sigma^2 = -\frac{1}{2} \tag{16}$$

Therefore, we can define the entropy of normal distribution as:

$$H(X) = \frac{1}{2}\log(2\pi\sigma^2) + \frac{1}{2} \tag{17}$$

## C  SKEWNESS

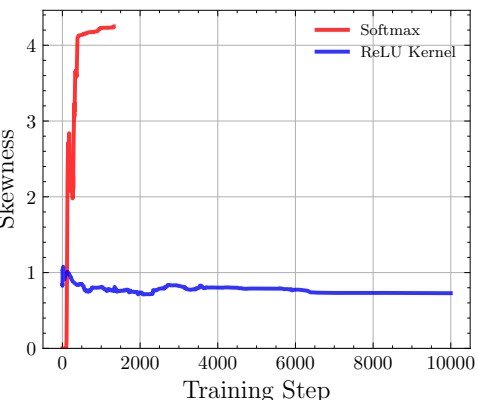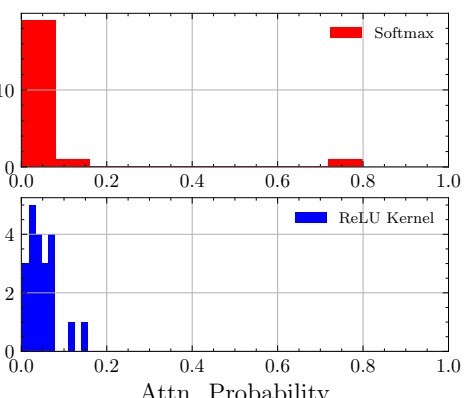

Figure 6: The figure on the left shows the average skewness of each row in the attention probabilities of the first layer. The left figure shows how the skewness of softmax-based self-attention and Lipschitz kernel self-attention changes during training. The figure on the right depicts the distribution of attention probabilities at the point where each method exhibits the highest skewness.

After passing through the softmax, values are represented between 0 and 1, and when the distribution becomes concentrated on a single value, entropy collapse occurs. If we consider each row of the attention weights as a probability distribution, this represents a highly imbalanced form. Skewness can be used as a measure of how much an activation function makes a distribution imbalanced. If attention weight's one of row vector is denoted by $\{a_1, a_2, a_3, ...a_N\}$, with the mean $\mu$ and standard deviation $\sigma$, the skewness is defined as follows:

$$S = \frac{1}{N}\sum_{i=1}^{N}\left(\frac{a_i - \mu^3}{\sigma}\right). \tag{18}$$

From the left plot in the 6, we can observe that the skewness of softmax rises sharply during training and reaches values close to the maximum skewness. This indicates that softmax is learning highly imbalanced distributions, where most tokens are attending to only one other token. In the top-right figure, we can see that most values are concentrated around 0 and fall below the mean, demonstrating strong positive skewness. In contrast, kernelized self-attention shows relatively lower skewness values, and from the bottom-right figure, we can see that values are more evenly distributed around the mean, indicating weaker positive skewness. Thus, in the case of the softmax function, the use of the exponential function amplifies specific values, further concentrating the distribution. On the other hand, kernelized self-attention, by utilizing kernels based on linearity, can learn a more evenly spread distribution.

# D LAYER-WISE ATTENTION ENTROPY

In Figure 1, we have shown that as the entropy of the first self-attention layer gradually decreases, the model's training becomes unstable. This is the experimental result showing the behavior of attention entropy in other layers.

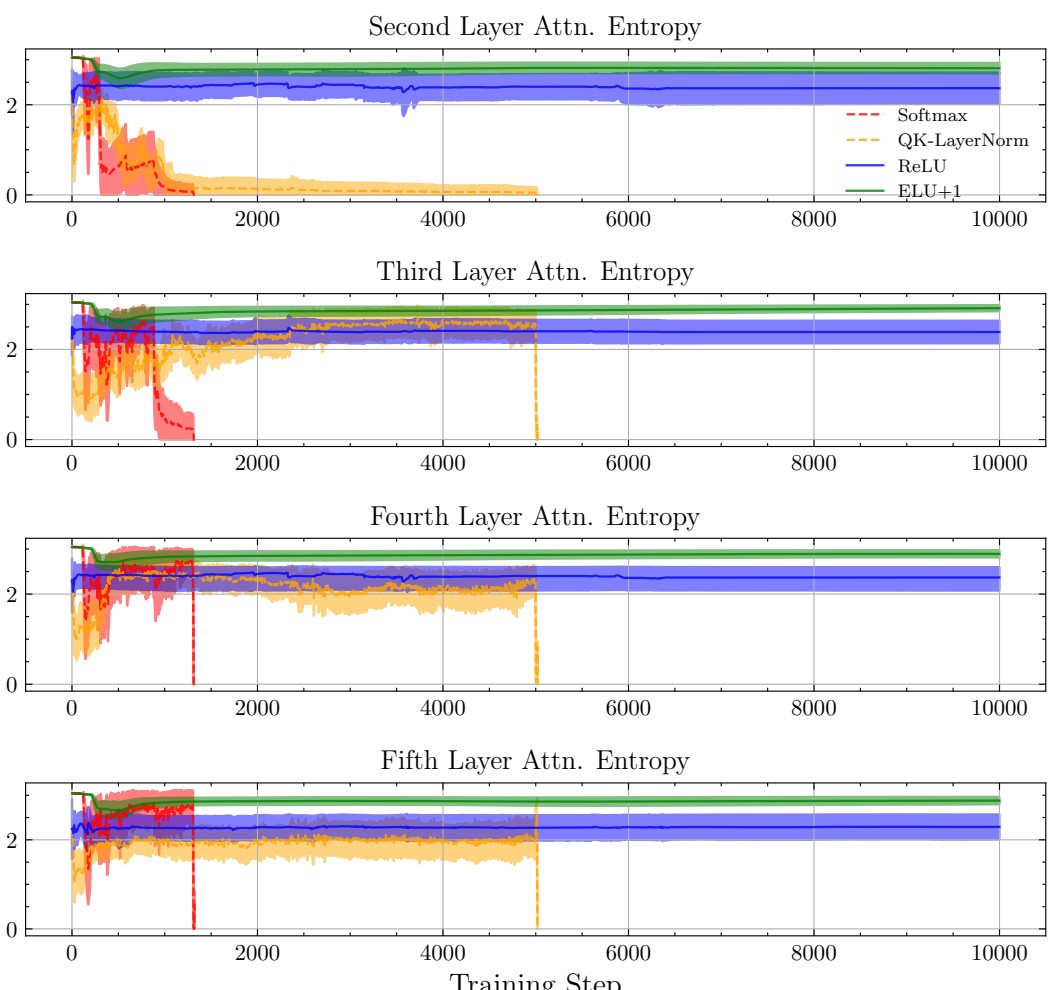

Figure 7: Attention entropy behavior across layers in softmax-based and Lipschitz kernel self-attention. Lipschitz kernel self-attention shows high attention entropy in all layers, while softmax-based self-attention has low entropy in early layers and high entropy in later layers.

