# OpenReview forum: "Transformer Training Instability of Softmax and Lipschitz-Kernel Attentions"
_ICLR.cc/2025/Conference — ICLR 2025 Conference Withdrawn Submission_

### Official Review · Reviewer_MZoD · 2024-10-26

**Soundness:** 1
**Presentation:** 2
**Contribution:** 1
**Rating:** 3
**Confidence:** 5

**Summary:**

The authors study the issue of attention entropy collapse in self-attention mechanisms that use the softmax activation function. The authors suggest that this issue is due to the fact that softmax attention is not Lipshitz, meaning a small perturbation of the input can lead to a large perturbation of the output in a non-controlled way. They then suggest that kernel based attention, which is Lipshitz, does not suffer from this problem. Thus they argue that Lipshitzness of the attention is the key role in keeping entropy from collapsing. Furthermore, they discuss how entropy collapse can lead to training instability and suggest that this is primarily due to attention probabilities concentrating into one hot vectors.

**Strengths:**

**Originality:** The authors take up the study of an important problem namely the entropy collapse of attention with a softmax activation. Their approach is original rooted in the idea that entropy collapse is arising due to the lack of Lipshitz property of attention with a softmax activation.

**Quality:** The authors perform a variety of controlled experiments showing that kernel based attention does not suffer from entropy collapse.

**Clarity:** The authors take the time to explain the problem of attention entropy collapse and clarify why this is an important problem.

 **Significance:** The authors provide arguments suggesting that entropy collapse is completely due to the lack of Lipshitzness of softmax attention which is significant as it suggest researchers should be designing transformers in a different manner.

**Weaknesses:**

**Novelty:** Unfortunately I feel the paper does not have any real novelty. The authors show that softmax attention suffers from entropy collapse and suggest that this is due to such attention mechanisms failing to be Lipshitz. Yet they only really provide empirical evidence of this and do not perform any sort of theoretical analysis which motivates their empirical work. They then suggest that kernel based attention is better because it is Lipshitz yet again do not provide any theoretical understanding that motivated their experiments. Furthermore, the authors don't seem to provide any analysis on current transformers used within the literature making it very difficult to see any real novelty in their work.

**Lack of theoretical motivation:** The paper has no real theoretical motivation. The authors carry out small scale experiments with ablations that suggest kernel based attention is better than the usual softmax based attention yet they do not motivate their arguments with theory in any way.

**Lack of experiments:** The authors seem to carry out experiments on a small scale transformer yet which transformer and what exact architecture seem to be never been discussed. They give some details in section 5.1 but these are not enough to give the readers a good understanding of what architecture they are using. I also looked at appendix A where the authors show their implementation details but I cannot see any details on what transformer they are working on. They say they are using a simply transformer in appendix A but no details are given. Furthermore, the authors  suggest that softmax attention suffers from entropy collapse and this leads to gradient norm exploding preventing model convergence (p. 2 second paragraph). However, many transformer models with softmax attention, such as Vision transformers [1], are known to converge so this seems to contradict what the authors are saying. They do not back this up with experiments that clearly show modern day transformers used in the community suffer from convergence issues.

[1] Steiner et al: How to train your ViT? Data, Augmentation, and Regularization in Vision Transformers, TMLR 2022.

**Questions:**

1). Could you please explain what type of transformer your experiments are carried out on? It is possible that I have missed the details but I couldn't find any details about this even in the appendix. You give some details in section 3.1 but these are not enough to give the reader insight into the exact transformer architectures you are using within your experiments.

2). Your main argument is based on the Lipshitzness of attention which is a quantitative measure of how the output of a function changes when its inputs are perturbed. Yet you don't give any theoretical argument as to how the lack of Lipshitzness leads to entropy collapse. Would you be able to give some theoretical insight as to how this connection comes about?

3). You mention on p. 2 paragraph 2 (line 062 to 063) that entropy collapse leads to gradient norm exploding which in turn prevents a transformer model using softmax attention to converge. Yet most transformers in the literature use softmax attention and are able to converge. If what you are saying is in fact true we should see that many transformers do not converge, however this is not what we see. Could you please explain this apparent contradiction with what you mention on p. 2 paragraph 2 (line 062 to 063) and the fact that in the literature many transformers use softmax activation? In fact, softmax is still one of the most used activations for attention.

4). On p.2, paragraph 2 you mention that because the softmax function is not Lipshitz continuous it leads to entropy collapse of the softmax attention. Your paper then suggests that kernel based attention is better as it is Lipshitz. Would I be right in then suggesting that as an alternative to softmax one should take an activation that is Lipshitz on the whole real line? Could softmax be replaced with any Lipshitz activation that is non-constant?

---

> ### Author Response · Authors · 2024-11-15
>
> *[W1] Novelty*
>
> - Our paper does not explicitly prove that softmax is non-Lipschitz, as this is a well-established fact due to the exponential nature of its formulation. It can be inferred from the kernel in Section 3.1 and Definition 3.1 (to be included in the Appendix). The claim that kernel-based attention outperforms softmax is supported by the fact that Lipschitz kernel self-attention prevents entropy collapse, enabling stable learning. The cause of such collapse is substantiated through *controlled experiments* examining *whether the function is sensitive to variance*. Comparisons with existing Transformers were conducted by testing against previous methods that attempted to prevent entropy collapse while still relying on softmax function.
>
> *[W2] Lack of theoretical motivation*
>
> - While the theoretical basis may be limited, we have provided clear evidence that the attention entropy patterns differ significantly between kernel attention based on Lipschitz functions and attention based on the softmax function. This difference is due to whether the function increases or does not increase the variance, which we have substantiated in our findings.
>
> *[W3] Lack of Experiments*
>
> - We acknowledge that the description of the architecture may be insufficient, and we plan to add further explanations in the Appendix. The architecture used in our paper follows the structure (Mahankali et al., 2023; Ahn et al., 2023; 2024), utilizing only the self-attention layer. This structure was chosen because it is well-suited for observing how attention entropy collapse in self-attention can lead to unstable training. Additionally, while Transformers or ViTs are known to converge well in some cases, this is only partially true; as model size increases (e.g., LLMs), sensitivity to the learning rate grows, and this phenomenon has been widely reported (Wang et al., 2021; Wortsman et al., 2023; Zhai et al., 2023; Dehghani et al., 2023).
>
>
> *[Q1] The architecture used in the experiments*
>
> - The Transformer architecture used in this study consists solely of stacked self-attention layers, with weights applied only to the query, key, and value. This structure was chosen to analyze whether attention entropy collapse occurring exclusively within self-attention impacts training stability.
>
> *[Q2] Theoretical insight of how the lack of Lipshitzness leads to entropy collapse*
>
> - We are planning to provide theoretical and experimental insights to establish a connection between Lipschitz continuity and entropy.
>
> *[Q3] How do you explain the fact that most transformers using softmax attention converge successfully?*
>
> - As you mentioned, Transformers, which are based on the self-attention mechanism, have demonstrated strong performance across various domains. However, with current research on Transformers, it has become well-known that as model size increases or with certain hyper-parameters (e.g., learning rate), these models become highly sensitive, making them challenging to optimize (Dehghani et al., 2023; Wortsman et al., 2023). Also, softmax function, which utilizes an exponential function, is also extremely sensitive to input changes, aligning with the fact that it is highly susceptible to variance (Wang et al. 2021; Shen et al., 2023; Hoffmann et al., 2023).
>
>
> *[Q4] Would I be right in then suggesting that as an alternative to softmax one should take an activation that is Lipshitz on the whole real line? Could softmax be replaced with any Lipshitz activation that is non-constant?*
>
> - The main argument presented in the paper is that by re-weighting attention logits with a Lipschitz continuous function, specifically one with a rate of change not exceeding 1, variance can be maintained stably, thus preventing collapse. Therefore, it may be beneficial to use activation functions that are Lipschitz continuous over the entire real range. However, considering that traditional self-attention uses softmax partly due to its non-negative output, it is preferable to use Lipschitz functions like ReLU and ELU+1 as kernels, as they respect this non-negativity characteristic.

---

> > ### Comment · Reviewer_MZoD · 2024-11-16
> > **Response by reviewer**
> >
> > Thanks for you response. Here I respond to what you have said.
> >
> > [W1]. I understand you have performed controlled experiments and I think there is no issue with that and it is great you did that. However, in order to show there is applicability of the deductions you make from your controlled experiment it would have been good to perform on analysis an large more common transformers people use within in the literature for applications. Such as a ViT for image classification say on the imagenet-1k dataset.
> >
> > [W2]. You have indeed provided evidence but only in a controlled case. You have not provided evidence to show your results from controlled experiments follow through for transformers people use in applications. It is one thing to say you get evidence for a controlled situation however what we as reviewers need to understand is whether that claim can be followed through to cases people actually use in the literature. If not, then the results you have provided through your controlled experiments are not useful to the community. This is where theory comes in. If you were to give a theoretical basis of your arguments we could then see how your methods could be applied to transformers people use within the literature.
> >
> > [W3]. Thank you. I will have a look at those citations.
> >
> > Q1. Thank you for clearing this up.
> >
> > Q2. Ok when you have updated your paper with the theoretical and experimental insights please let me know and I will take a look.
> >
> >  Q3. So does this mean that Lipshitz-kernel attention in a transformer can be trained with larger learning rates than softmax?
> >
> > Q4. Thank you. So would this mean activation functions like ReLU and ELU + 1 should replace softmax as their are Lipshitz and therefore won't lead to collapse? Also, would they perform better than softmax?

---

> > > ### Author Response · Authors · 2024-11-19
> > >
> > > *[W1] Controlled experiment it would have been good to perform on analysis an large more common transformers*
> > >
> > > - We fully understand the need for experimental results on this. However, there is already sufficient empirical evidence from previous approaches studying entropy collapse, showing that such limitations arise in practical Transformer models (Wortsman et al., 2023). Nevertheless, we are planning to conduct experiments on commonly used, general Transformer models in the near future.
> > >
> > >
> > > *[W2] Controlled experiments follow through for transformers people use in applications*
> > >
> > > - We fully understand the need for experiments on practical Transformer models, and we will present the results from such experiments as evidence to support our claims.
> > >
> > >
> > > *[Q3] So does this mean that Lipshitz-kernel attention in a transformer can be trained with larger learning rates than softmax?*
> > >
> > > - The purpose of our experiments with high learning rates was to approximate LLMs by leveraging smaller model sizes and higher learning rates (Wortsman et al., 2023, Zhai et al., 2023) . Although we did not include the attention entropy and training instability results under varying learning rates in the appendix, we observed that softmax is highly sensitive, whereas Lipschitz kernels demonstrated significant robustness. In conclusion, in LLM settings, softmax re-weighting exhibits highly unstable entropy and is sensitive to learning rates. In contrast, Lipschitz kernels show stable attention entropy patterns even in LLM settings and are not sensitive to learning rates.
> > >
> > > *[Q4]  So would this mean activation functions like ReLU and ELU + 1 should replace softmax as their are Lipshitz and therefore won't lead to collapse? Also, would they perform better than softmax?*
> > >
> > > -  From a performance perspective, further research is needed, but it is evident that the attention entropy of softmax is unstable. Therefore, replacing softmax with a Lipschitz continuous function could be a desirable choice to address learning instability.

---

> ### Comment · Reviewer_MZoD · 2024-11-25
> **Response by author**
>
> Thank you for your response.
>
> I am still not convinced with the authors' work. While interesting I still feel there is still significant areas that need working which I have outlined previously and don't feel the reviewers have adequately addressed. I have gone back and looked at the paper and noticed the authors have not made any real changes that they said they would in response to my initial review. There is still a significant lack of theoretical motivation/contribution as well as a lack of experiments on transformers that are actually used by researchers within the field.
>
> Due to this I will keep my original rating of the paper and thank the authors for the discussion.

---

### Official Review · Reviewer_1WXL · 2024-10-27

**Soundness:** 2
**Presentation:** 2
**Contribution:** 1
**Rating:** 3
**Confidence:** 5

**Summary:**

This paper argues that the underlying reason why the attention entropy collapse leads to the training instability and then proposes Lipschitz-kernel-based attention which is able to prevent attention entropy collapse.

**Strengths:**

**Originality:** Introducing Lipschitz Kernel Attention is interesting.

**Quality:** This paper provides mathematical explanations and experiments showing the proposed method having more stable attention entropy.

**Clarity:** Section 3 is well organized and it is clear to follow the logic in general.

**Significance:** Reducing attention complexity is promising for large scale models. Attention mechanism is not understood well and understanding attention through entropy is appreciated and interesting.

**Weaknesses:**

**Novelty:** The method authors proposed in Eq. 5 is not new [1] [2]. The mathematical part in Section 3.5, which is used to support the argument, has also been introduced in other papers(Line 417).

**Experiment Setting:** I am confused with the settings authors choose. The experiment authors conduct is on a Simple Transformer with SGD, very high learning rate(5e-1), low hidden dimension(3), low attention heads(1), and many others. It would be good to have experiments on standard or larger transformer architecture, such as [3].

[1] Efficientvit: Multi-scale linear attention for high-resolution dense prediction. Han Cai, Junyan Li, Muyan Hu, Chuang Gan, and Song Han.  ICCV 2023

[2] cosformer: Rethinking softmax in attention. Zhen Qin, Weixuan Sun, Hui Deng, Dongxu Li, Yunshen Wei, Baohong Lv, Junjie Yan, Lingpeng Kong, and Yiran Zhong. ICLR 2022

[3] Attention is all you need. A Vaswani, N Shazeer, N Parmar, J Uszkoreit, L Jones, AN Gomez, Ł Kaiser, I Polosukhin. NIPS 2017

**Questions:**

1. Line 50 "Previous approaches primarily focused on interpreting the results" is not clear and it would be good to clarify the reference nad which approaches you are referring to

2. Line 148 why is that "we can reduce quadratic time complexity into $O(ND^2) = O(N)$ with respect to the input sequence length $N$ as the hidden dimension is smaller than sequence length

---

> ### Author Response · Authors · 2024-11-15
>
> *[W1] Novelty*
>
> - What we are proposing in this paper is not so much a method, but rather an insight into how entropy collapse—a limitation that can arise in attention modules—occurs due to the non-Lipschitz nature of softmax. We provide empirical evidence that replacing softmax with other Lipschitz continuous functions in kernel-based self-attention, aimed at reducing computation, more effectively maintains stable entropy. Furthermore, *we theoretically and experimentally investigate how the Lipschitz property impacts entropy and why entropy collapse leads to training instability*.
>
> *[W2] Experiment Setting*
>
> - The purpose of using a high learning rate with a small model scale is to approximate a large language model setting, as done in (Wortsman et al. 2023, Von Oswald et al., 2023; Ahn et al., 2023; 2024). Our experiments demonstrate that even under these conditions, entropy collapse does not occur, and stable training is achieved with Lipschitz continuous self-attention. While previous studies have extensively experimented with standard Transformer scales, our work focuses specifically on attention entropy collapse using only self-attention layers. By employing a small scale model and a high learning rate, we approximate an LLM setting and conduct controlled experiments to analyze the causes of this issue.
>
> *[Q1] Line 50 "Previous approaches primarily focused on interpreting the results" is not clear and it would be good to clarify the reference.*
>
> - It has been noted that the explanation of related works is insufficient, and we plan to further supplement and expand this section.
>
> *[Q2] Line 148 why is that "we can reduce quadratic time complexity into $O(ND^2)=O(N)$ with respect to the input sequence length $N$ as the hidden dimension is smaller than sequence length.*
>
> - In this paper, the Lipschitz self-attention used approximates the traditional kernel-based self-attention by replacing the softmax function with a kernel function, as shown in eq (4). In conventional self-attention, as seen in eq (1), calculating each row of the $N \times N$ attention probabilities involves first taking the dot product of queries and keys, resulting in a computational complexity of $O(N^2D)$. By restructuring the equation using the associative property, as shown in eq (5), the method computes the product of keys and values first, reducing the complexity to $O(D^2N)$. In our experiments, instead of $\phi$, we employed Lipschitz functions (e.g., ReLU or ELU+1) to observe performance. The assumption that the hidden dimension is smaller than the sequence length $N$ aligns with practices in traditional NLP domains, where the *sequence length is generally larger than the hidden dimension, leading to reduced computational complexity (as referenced in Qin et al., 2022)*.

---

### Official Review · Reviewer_amhu · 2024-10-30

**Soundness:** 1
**Presentation:** 2
**Contribution:** 1
**Rating:** 1
**Confidence:** 5

**Summary:**

This paper describes and reaffirms the connection between entropy collapse, a phenomenon recognized in the literature see (Zhai et al., 2023)—and the activation function used in attention layers, comparing row-wise softmax with Lipschitz functions like ReLU or ELU.
It essentially rephrases already known results in the literature, providing a light high level understanding of the problem. The provided experiments are limited in scale and validate existing results, without presenting any new theoretical contributions.

**Strengths:**

The research question the authors tried to address is of high importance and relevance to the Machine Learning community, with a lot of significant direct implications at stake.

**Weaknesses:**

1. This submission overall reads more like a report than a research paper due to a significant lack of novelty. Previous works have extensively explored the replacement of softmax with the ReLU activation in both practical (Hron et al., 2020) and theoretical contexts (Bai et al., 2023; Fu et al., 2023). Wortsman et al. (2023a) replace softmax with many normalized different variants. Ramapuram et al. 2024 investigate sigmoid activation function. Additionally, Zhai et al., 2023 directly connect softmax layers to entropy collapse, leaving this submission with little to no new insight to contribute.

2. The manuscript appears unpolished and difficult to read. Below are a few specific issues among the many present:

- line 40: a verb is missing
- line 51: word "bridge" shouldn’t be there
- lines 128 and 133: the quadratic bottleneck issue is repeated twice
- lines 55, 138, 142, etc.: articles ("the" or "a") are often missing
- line 112 is not a complete sentence and should be separated from the rest by a comma
- lines 114 to 117, lines 151-154 are unreadable
- over the whole manuscript: past and present tenses are inconsistently used to describe the literature
- line 205: "that it can learn" -> "it can learn"

3. Figure 1.a should be presented on another scale for better visibility.

4. Definition 3.1 is redundant given the formal definition provided right before in the paragraph.

5. Definition 3.2 does not make mathematical sense as the left hand side, which is not a function of j, is indexed by j.

6. Plots would be ideally averaged over multiple runs.

7. The primary focus of the paper which is to explain why Lipschitz alternatives are to be preferred to softmax remains speculative: line 227: "To compare with the softmax-based attention, we experiment with a kernelized self-attention with a lipschitz kernel function […], which we expect to mitigate entropy collase".

8. The only mathematical assertion in this paper is questionable and not convincing. Line 419 being an inequality, an increase of the upper bound does not necessarily indicate an increase in the original quantity. The bound can simply become vacuous and uninformative as opposed to the authors’claim (line 430): "the norm of the attention probabilities increases", which is therefore unsupported.

9. Minor comment: Introducing the variance as in definition 3.2 is a good idea but it essentially reduces to considering a temperature within softmax, the latter being analysed in Zhai et al., 2023.

**Questions:**

In its current form, this submission is not suitable as a research paper. However, I encourage the authors to consult the following papers for insights into how the Lipschitz property of the attention layer influences training dynamics (Ramapuram et al. 2024, Jha et al. 2024). This may offer useful directions if they wish to further pursue this topic.

---

> ### Author Response · Authors · 2024-11-15
>
> *[W1] The differences between our work and other studies that replace softmax with ReLU or sigmoid functions, as well as those focusing on attention entropy collapse.*
>
> - We theoretically and experimentally connect the causes of attention entropy collapse and the resulting training instability by comparing non-Lipschitz and Lipschitz functions of re-weighting attention logits. Therefore, it is not limited to the ReLU function but is used as a representative example of a Lipschitz function that ensures non-negative values. Among the related works, Hron et al. use the ReLU function because it guarantees non-negative values like softmax and behaves similarly to the identity function. Wortsman et al. concentrate on demonstrating that ReLU outperforms other activation functions, while Bai et al., 2023, Fu et al., 2023 use ReLU purely for theoretical convenience without additional reasoning.
> - Sigmoid Attention (Ramapuram et al. 2024), similar to our work, leverages the Lipschitz continuity of the sigmoid function for stability and calculates similarity scores independently, resulting in more gradual output changes than softmax. However, these studies lack empirical evidence on how stability differs between the two functions, focusing instead on computational experiments.
> - Zhai (et al. 2023) investigate entropy in softmax-based self-attention layers, providing a lower bound for entropy and highlighting the growing weights of query and key as a primary cause. But, our research focuses on how the Lipschitz property of re-weighting functions affects the behavior of attention entropy and systematically conducts experiments to establish its step-by-step connection to training instability.
> - In this paper, we show that, unlike softmax, which increases the variance of attention logits, *Lipschitz functions maintain stable variance*, *helping to prevent entropy collapse*. We also provide *theoretical and empirical evidence linking attention entropy collapse to training instability*. This forms the novelty of our research.
>
>
> *[W2-W5] Revised the content of the paper*
>
> - That section has been fully revised.
>
>
> *[W6] Plots would be ideally averaged over multiple runs.*
>
> - In the experimental results shown in the plot, for non-Lipschitz functions like softmax or QK-LayerNorm, the gradient norm became excessively large, ultimately causing training to halt. Therefore, we believe that averaging over multiple runs is not appropriate in this case.
>
> *[W7] Why Lipschitz alternatives are to be preferred to softmax remains speculative.*
>
> - The Lipschitz property of a re-weighting function that maps the inner product of queries and keys to values between 0 and 1 can influence attention entropy collapse. Using *non-Lipschitz functions with high rates of change significantly increases the variance of attention logits*, leading to one-hot-like distributions that focus excessively on a single token. In contrast, when re-weighting with Lipschitz functions, even with large variance, *the rate of change remains bounded within a constant, resulting in more balanced distributions*. We demonstrate this robustness against entropy collapse both theoretically and experimentally.
>
> *[W8] Line 419 being an inequality, an increase of the upper bound does not necessarily indicate an increase in the original quantity.*
>
>  - The first term in the upper bound, $\||P\||_F$​, represents the norm of the attention probabilities. This norm reaches its maximum value when attention entropy collapses completely (i.e., when all vectors along the sequence axis become one-hot). Therefore, *when attention entropy collapses*—resulting in a *larger $\||P\||_F$*—the upper bound increases, leading to training instability. Conversely, as $\||P\||_F$​ decreases, the upper bound lowers, also contributing to stability.
>
> *[W9] Considering a temperature within softmax*
>
> - Temperature adjustment does not make softmax a Lipschitz function or remove its non-Lipschitz property. It simply adjusts how sensitive softmax is to changes in its logits.
>
> *[Q1] In its current form, this submission is not suitable as a research paper. However, I encourage the authors to consult the following papers for insights into how the Lipschitz property of the attention layer influences training dynamics (Ramapuram et al. 2024; Jha et al. 2024). This may offer useful directions if they wish to further pursue this topic*
>
> - Thank you for encouraging our research. We will review the referenced paper as well as additional related studies.

---

> > ### Comment · Reviewer_amhu · 2024-11-24
> >
> > I thank the authors for their answers.
> >
> > W1. I do not acknowledge the theoretical contribution that the authors claim.
> >
> > W6. True, maybe showing multiple runs in the appendix would do the work?
> >
> > W7. The authors claim showing their claim 'theoretically', where can I find the proof? It appears that the high-level arguments presented are mistaken for theoretical proofs.
> >
> > W8. I maintain that having an "upper bound that increases" does not provide any meaningful theoretical insights.
> >
> > W9. It is simply not true. The authors might consider looking into the literature, e.g. https://arxiv.org/abs/1704.00805
> >
> > I hope the authors find my feedback helpful. In light of this discussion, I maintain my score.

---

> ### Author Response · Authors · 2024-11-25
>
> *[W1] I do not acknowledge the theoretical contribution that the authors claim.*
>
> - We are supplementing the theoretical contribution.
>
>
> *[W6] True, maybe showing multiple runs in the appendix would do the work?*
>
> - We will supplement the result and include it in the appendix.
>
> *[W6] The authors claim showing their claim 'theoretically', where can I find the proof? It appears that the high-level arguments presented are mistaken for theoretical proofs.*
>
> -  It is clear that using the softmax function within self-attention can lead to a reduction in attention entropy, potentially resulting in collapse and contributing to training instability. However, the theoretical foundation for this explanation may be insufficient, and further attempts at a more detailed theoretical proof are underway. Nonetheless, we believe that the experimental results sufficiently support our claims.
>
>
> *[W8] I maintain that having an "upper bound that increases" does not provide any meaningful theoretical insights.*
>
> - While the increasing upper bound itself may lack theoretical insights, we have supplemented it with empirical evidence. Specifically, as shown in Figure 5, for softmax-based self-attention, the norm of P increased as the attention entropy decreased. In contrast, for Lipschitz kernel self-attention, stability was observed.
>
> *[W9] It is simply not true. The authors might consider looking into the literature, e.g. https://arxiv.org/abs/1704.00805*
>
> - The paper you introduced confirms that the softmax function is indeed Lipschitz continuous. However, in self-attention, where $X$ is the input and $P(X)$ represents the attention probabilities obtained after applying the softmax function, the multiplication of $P(X)$ and $X$ in the self-attention mechanism causes the overall mapping to lose Lipschitz continuity. Additionally, since the softmax function is highly sensitive to its input values, when $F(X)=P(X)X$ (self-attention function that dot-product the attention probabilities with the values), the rate of change of this F(X) function is greatly influenced by the saturation of the softmax, which varies depending on the input values (https://arxiv.org/pdf/2006.04710).

---

### Official Review · Reviewer_jUeM · 2024-11-03

**Soundness:** 2
**Presentation:** 3
**Contribution:** 2
**Rating:** 3
**Confidence:** 5

**Summary:**

This paper addresses the problem of entropy collapse in self-attention-based transformer models, which leads to training instability in such networks. The authors show that softmax-based self-attention suffers from attention entropy collapse due to non-Lipschitz property and can thus be mitigated using Lipschitz kernel-based attention.

**Strengths:**

1. The paper presentation is clear and easy to follow.

2. The issue of training instability in self-attention-based transformers is an interesting problem for understanding the inner workings of transformer models.

**Weaknesses:**

1. The paper does not mention the data/model setting under which experiments have been performed. Details given in section 3.1 are not enough to reproduce the results, the setup section needs more details to better understand the observations in the paper.

2. It is not clear why the concentration of attention on a few tokens should lead to instability, as the premise of attention is to give higher attention to the relevant tokens and ignore irrelevant tokens for specific downstream tasks.

3. Under what setting have the experiments shown in Figure 2 been performed? e.g. share the details regarding the training data and how it is sampled, these details are missing from the paper

4. Apart from stable entropy, is there a specific reason that leads to stable training for Lipschitz-Kernel attention as compared to softmax-based attention?

**Questions:**

Questions to authors:

1. The paper does not mention the data/model setting under which experiments have been performed.

2. It is not clear why the concentration of attention on a few tokens should lead to instability, as the premise of attention is to give higher attention to the relevant tokens and ignore irrelevant tokens for specific downstream tasks. Authors should include a detailed analysis of  how their observation relates to the intended function of attention in different types of tasks.

3. Under what setting have the experiments shown in Figure 2 been performed? Please clarify. e.g. share the details regarding the training data and how it is sampled, these details are missing from the paper

4. Apart from stable entropy, is there a specific reason that leads to stable training for Lipschitz-Kernel attention as compared to softmax-based attention?

---

> ### Author Response · Authors · 2024-11-15
>
> Thank you for your response and for sharing the alternative resources.
>
>
> *[W1] Data/Model Setting*
>
> - We have specified the experimental settings in Appendix A. Unlike previous studies that analyze attention entropy collapse in larger-scale models with downstream tasks, we focused on this attention entropy collapse with the only attention layer to clearly demonstrate why this issue occurs and how this phenomenon can lead to training instability. Also, we conducted our experiments in a simpler linear regression task, thereby proving that this phenomenon can occur universally.
>
> *[W2] Why the concentration of attention on a few tokens should lead to instability?*
>
> - Attention entropy collapse occurs when the query-key inner product (also called the attention logits) is converted into an attention probability vector, leading to an excessive focus on a single token (Figure 2). At this point, if all vectors along the sequence dimension take the form of *one-hot vectors, the norm of the attention probabilities increases, causing the gradient to explode* and resulting in unstable training (line 423-430).
>
> *[W3] Under what setting have the experiments shown in Figure 2 been performed?*
>
> - Following the similar approach as in (Garg et al., 2022; Zhang et al., 2023; Mahankali et al., 2023; Von Oswald et al., 2023; Ahn et al., 2023; 2024), $x_i$ and $w_i$ were randomly sampled, $x, w \sim N(0, I)$ (It was mentioned on line 209). The $y_i$ was then calculated as the inner product of the two vectors, training the model on the linear regression $y_i=w^{\top}x_i$.
>
> *[W4] Apart from stable entropy, is there a specific reason that leads to stable training for Lipschitz-Kernel attention as compared to softmax-based attention?*
>
> - Our paper focuses on empirical evidence showing that training exclusively with a model composed solely of self-attention layers leads to unstable training when using non-Lipschitz functions (such as softmax and QK-LayerNorm). By concentrating on how Lipschitz continuous functions prevent this instability, other issues were unlikely to arise.
>
> *[Q1] Data/Model Setting*
>
> - In Appendix A, we detail the experimental settings, focusing on attention entropy collapse using only the attention layer to clearly demonstrate its causes and impact on training instability. By conducting experiments on a simpler linear regression task, we show that this phenomenon can occur universally, unlike prior studies that focus on larger-scale models with downstream tasks.
>
> *[Q2] Why the concentration of attention on a few tokens should lead to instability and an analysis of how their observations align with attention's purpose in different task.*
>
> - When attention is concentrated on a few tokens, meaning that attention entropy decreases and collapses, leading to training instability, we provide evidence that this results in an increase in the norm of the attention probabilities, causing gradient exploding (as mentioned in Section 3.5). Additionally, attention entropy collapse across various tasks has already been explored in previous studies (Zhai et al., 2023; Shen et al., 2023; Jiang et al., 2023). These experiments were conducted on models and tasks of a scale commonly used in practical applications. In contrast, our paper conducted experiments in a more controlled setting to analyze why entropy collapse occurs and how it leads to training instability.
>
>
> *[Q3] Under what setting have the experiments shown in Figure 2 been performed?*
> - As explained in the response to W2, the data was sampled accordingly, and the experimental settings are provided in the Appendix A.
>
>
> *[Q4] Is there a specific reason that leads to stable training for Lipschitz-Kernel attention as compared to softmax-based attention?*
>
> - We focused on the attention entropy collapse occurring within self-attention and accordingly used an architecture composed solely of self-attention layers. Therefore, we cannot determine if other factors outside of attention play a role in this collapse.

---

> > ### Comment · Reviewer_jUeM · 2024-11-27
> > **Reply to Official Comments by Author**
> >
> > I thank the authors for their answers.
> >
> > [W1] I have gone through the data settings explained in the main paper/appendix, it needs more details as it is not possible to reproduce the experiments just by following the paper.
> >
> > [W2] The explanation does not clarify why attention entropy collapse occurs as the premise of attention is based on the fact that we want to select a few tokens among all the tokens in a sequence for contextual representation of the query token.
> >
> > [W3] Please provide more details for the experimental setting.
> >
> > [W4] I think my question was not clear, what I wanted ask was in the setting which author considers is there any other property except stable entropy which can also influence the training stability.
> >
> > Based on the author's response, and other reviewer's comments, I will maintain my score.

---

### Note · Authors · 2024-12-02

**Comment:**

We would like to withdraw our submission as we plan to submit it to another conference. Thank you for taking the time to review our work.

**Withdrawal Confirmation:**

I have read and agree with the venue's withdrawal policy on behalf of myself and my co-authors.